# Expression of Fibrosis-Related Genes in Liver and Kidney Fibrosis in Comparison to Inflammatory Bowel Diseases

**DOI:** 10.3390/cells11030314

**Published:** 2022-01-18

**Authors:** Miha Jerala, Nina Hauptman, Nika Kojc, Nina Zidar

**Affiliations:** Institute of Pathology, Faculty of Medicine, University of Ljubljana, Korytkova 2, 1000 Ljubljana, Slovenia; miha.jerala@mf.uni-lj.si (M.J.); nina.hauptman@mf.uni-lj.si (N.H.); nika.kojc@mf.uni-lj.si (N.K.)

**Keywords:** inflammatory bowel diseases, fibrosis, Crohn’s disease, ulcerative colitis, gene expression

## Abstract

Fibrosis is an important feature of inflammatory bowel diseases (IBD), but its pathogenesis is incompletely understood. Our aim was to identify genes important for fibrosis in IBD by comparison with kidney and liver fibrosis. First, we performed bioinformatics analysis of Gene Expression Omnibus datasets of liver and kidney fibrosis and identified *CXCL9, THBS2, MGP, PTPRC, CD52, GZMA, DPT* and *DCN* as potentially important genes with altered expression in fibrosis. We then performed qPCR analysis of the selected genes’ expression on samples of fibrotic kidney, liver, Crohn’s disease (CD) with and without fibrosis and ulcerative colitis (UC), in comparison to corresponding normal tissue. We found significantly altered expression in fibrosis for all selected genes. A significant difference for some genes was observed in CD with fibrosis in comparison to CD without fibrosis and UC. We conclude that similar changes in the expression of selected genes in liver, kidney fibrosis and IBD provide further evidence that fibrosis in IBD might share common mechanisms with other organs, supporting the hypothesis that fibrosis is the common pathway in diseases of various organs. Some genes were already active in IBD with inflammation without fibrosis, suggesting the early activation of profibrotic pathways or overlapping function in fibrosis and inflammation.

## 1. Introduction

Intestinal fibrosis is a common feature in inflammatory bowel diseases (IBD), particularly in Crohn’s disease (CD) [1,2]. In CD, fibrosis with intestinal strictures develops in up to 70% of patients [3]. Despite significant progress in the treatment of CD with anti-inflammatory agents, there is little impact on the incidence of fibrosis and intestinal strictures that necessitate surgical intervention [2,4,5,6]. Fibrosis is commonly observed in ulcerative colitis (UC) as well, but it is limited to the submucosal layer and usually does not cause strictures, making it clinically less important [7,8,9]. Despite intensive studies of fibrosis in IBD, our understanding is still incomplete, resulting in the lack of effective therapies that would target fibrosis and prevent the formation of intestinal strictures and the need for surgical intervention [1,10].

Fibrosis is not unique to the bowel; it occurs in many organs, including the liver, kidney, heart, skin and lung. It is the end result of tissue damage and chronic injury, regardless of the involved organ [11]. Extensive research has been devoted to organ fibrosis in recent decades, with many similarities discovered in various organs. The major effector cell is the activated myofibroblast, which produces abundant amounts of extracellular matrix (ECM). There are many postulated origins for activated myofibroblasts, including transdifferentiation from resident cells or from circulating cells [11,12]. Recently, pericytes and hepatic stellate cells that function as specialized liver pericytes have come into focus as the most important source of myofibroblasts [13,14]. At the molecular level, there are many common mediators and signaling pathways involved in fibrosis, the most important being the transforming growth factor β (TGF-β) pathway, platelet-derived growth factor receptors (PDGFR) pathway, connective tissue growth factor (CTGF) pathway and others [11]. Fibrosis is generally regarded to be irreversible and permanent. However, there is evidence that it can be reversed to some degree, for example liver fibrosis, which can be reversed after cessation of the starting insult (e.g., treatment of infection, autoimmune disease, cessation of alcohol consumption) [14,15]. The reversal is thought to function through myofibroblast senescence and apoptosis, the clearance of excessive ECM by macrophages and matrix metalloproteinases, and the regeneration of functional hepatocytes [14,15,16]. It has been shown that fibrosis is a dynamic process with constant deposition and resorption of ECM occurring even in established fibrosis [17,18]. There is some evidence of the reversal of fibrosis in IBD after strictureplasty; however, its clinical importance remains to be determined [18,19].

We believe that we can improve our understanding of fibrosis in IBD by searching for processes that are common to fibrosis in other organs. The same processes should have a role in intestinal fibrosis as well. In our study, we used a bioinformatics approach to identify gene expression changes, common to fibrosis in the liver and kidney. We then analyzed these genes in biopsy samples of fibrotic liver and kidney in comparison to the corresponding normal tissue. Finally, we examined samples of CD, UC and normal colon to see whether any of these genes are altered in intestinal fibrosis as well.

## 2. Materials and Methods

### 2.1. Bioinformatics Analysis

In the bioinformatics analysis, we used freely available data from Gene Expression Omnibus (GEO), specifically GSE7392 and GSE84044, for studying fibrosis in kidney and liver, respectively. In kidney dataset GSE7392, we used 16 samples from 8 patients that developed subclinical interstitial fibrosis in renal allografts post-transplantation. In liver dataset GSE84044, 28 samples with Scheuer score 3 and 4 were used and 43 samples with Scheuer score 0, all from patients with chronic hepatitis B. Both datasets were created on the same platform, Affymetrix Human Genome U133 Plus 2.0 Array. 

Both projects’ raw data were downloaded and further manipulated separately in R programming language. Using the robust multichip average function in R package “affy”, the CEL files were converted into expression data [20]. After data normalization, gene filter was used to remove probes that had an intensity less than 100 in more than 20% of samples in each project. We used dendrograms in each project to verify that the two groups of samples did not overlap. Differentially expressed genes (DEG) were identified on the probe level using limma package in R for each individual project [21]. For constructed comparison (fibrosis compared with normal), the cut-off value was set as *p* ≤ 0.05. 

The set of DEGs was used to construct a protein–protein interaction (PPI) network. The PPI network was constructed with STRING v11 under the cut-off interaction score of 0.4 and with active interaction sources, such as Textmining, Experiments, Databases, Co-expression, Neighborhood, Gene Fusion, Co-occurrence (https://string-db.org/, accessed on 3 December 2020). 

### 2.2. Selected Gene Expression Analysis

#### 2.2.1. Patients’ Selection

Fibrosis-related gene analysis was performed on biopsy samples from 78 patients. There were:
11 patients with kidney fibrosis due to diabetic nephropathy and 9 patients as controls without fibrosis, with thin glomerular basement membrane, with no other abnormalities;10 patients with liver cirrhosis and 7 deceased patients with morphologically normal liver (pre-transplantation screening biopsies of potential donor livers);20 patients with CD of the colon (14 samples of colon with fibrosis, 14 samples of colon with inflammation without fibrosis), 10 patients with UC and 11 patients with colorectal carcinoma (samples of morphologically normal colon, taken from uninvolved margins of resection specimens).


#### 2.2.2. Tissue Acquisition

Tissue samples were obtained from diagnostic needle biopsies of the kidney and liver, and from surgical resections of the colon (CD, UC and colorectal cancer). Tissue samples were fixed in 10% buffered formalin for 24 h, prior to embedding in paraffin. After fixation, sections were cut and stained for routine histopathological examination. For the present study, we reviewed the original slides and retrieved the corresponding paraffin blocks from the archive of the Institute of Pathology, Faculty of Medicine, University of Ljubljana.

#### 2.2.3. RNA Isolation

For RNA isolation, three 10 µm-thick sections were cut from the archival paraffin blocks. Total RNA was extracted with MagMAX FFPE DNA/RNA Ultra Kit (Thermo Fisher Scientific, Waltham, MA, USA), according to the manufacturers’ protocol. The extracted RNA concentration and quality was assessed using a NanoDrop 1000 spectrophotometer (Thermo Fisher Scientific) and Qubit Fluorimetric Quantification (Thermo Fisher Scientific).

#### 2.2.4. Reverse Transcription and Quantitative PCR

Reverse transcription (RT) and quantitative PCR (qPCR) were performed to measure the level of gene expression of *MGP*, *THBS2*, *CXCL9*, *CD52*, *DPT*, *GZMA*, *PTPRC* or *DCN*, while *B2M* and *IPO8* were used as reference genes. The extracted RNA was reverse transcribed using OneTaq^®^ RT-PCR Kit (New England BioLabs, Ipswich, MA, USA), according to the manufacturers’ protocol. Separate pools of RNA samples were created from samples of each tissue group for efficiency test. After RT, cDNA of pooled samples was diluted 5- to 625-fold and the PCR probes were tested for qPCR efficiency. All reactions assessing efficiency were performed in triplicate on Rotor-Gene Q system (Qiagen GmbH, Hilden, Germany). All qPCR reactions were performed in duplicate.

#### 2.2.5. Statistical Analysis of qPCR Experiments

To calculate relative gene expression, Cq values were corrected as described by Latham. ΔCq was obtained by deducting the geometric mean of the Cq values of reference genes (*B2M* and *IPO8*) from the Cq of the gene of interest (*MGP*, *THBS2*, *CXCL9*, *CD52*, *DPT*, *GZMA, PTPRC* or *DCN*). Differences in expression between fibrotic tissue and its corresponding normal control were analyzed for significance using the Mann–Whitney test. All statistical analyses and correlations of gene expression within each sample group were calculated in SPSS version 27 (IBM Corp., Armonk, NY, USA), with a cut-off value of *p* ≤ 0.05.

## 3. Results

### 3.1. Bioinformatics Analysis

The results of the bioinformatics analysis revealed 57 genes which had a significantly altered gene expression between fibrotic and normal tissue in kidney and liver (Appendix A). The majority of those genes are members of the groups collagen, immunoglobulin heavy or histocompatibility complex. 

The PPI network (Appendix A, Figure 1) clearly shows two groups of linked proteins, at the center of which are *PTPRC* on one side and *COL1A1* on the other side. The PPI network was the basis for the selection of genes for further experimental validation, where eight genes were selected. Four genes with a direct interaction with *COL1A1, MGP, THBS2, DCN* and *DPT*, were part of or connected to extracellular matrix and its organization, while on the other side we selected *PTPRC, CD52, GZMA* and *CXCL9*, which are involved in the immune response through different regulation mechanisms of T cells. All the selected genes were significantly up-regulated in fibrosis (Table 1).

### 3.2. Validation of Changes in Gene Expression

#### 3.2.1. Patient Population

The most important demographic and clinical data are presented in Table 2.

#### 3.2.2. Expression of Selected Fibrosis-Related Genes in the Liver and Kidney

In liver cirrhosis as compared with normal liver, genes *THBS2, MGP, PTPRC, CD52* and *DPT* (ΔΔCq = 1.46, *p* = 0.03; ΔΔCq = 3.43, *p* < 0.01; ΔΔCq = 3.06, *p* < 0.01; ΔΔCq = 2.39, *p* < 0.01; ΔΔCq = 4.68, *p* < 0.001, respectively) had significantly up-regulated expression, while the expression of genes *CXCL9*, *GZMA* and *DCN* (ΔΔCq = 0.76, *p* = 0.21; ΔΔCq = 0.31, *p* = 0.31; ΔΔCq = 0.81, *p* = 0.056, respectively) was not significantly altered (Figure 2, Table 3).

Of the eight chosen genes, the following six, *CXCL9*, *THBS2*, *MGP*, *PTPRC*, *CD52* and *DCN* (ΔΔCq = 3.10, *p* < 0.001; ΔΔCq = 2.11, *p* < 0.001; ΔΔCq = 1.70, *p* = 0.012; ΔΔCq = 2.60, *p* < 0.001; ΔΔCq = 3.83, *p* < 0.001; ΔΔCq = 1.45, *p* < 0.001, respectively), had significantly up-regulated expression in kidney fibrosis compared with normal kidney. *GZMA* and *DPT* (ΔΔCq = −0.21, *p* = 0.635; ΔΔCq = −1.31, *p* = 0.08, respectively) did not have significantly altered expression.

#### 3.2.3. Expression of Selected Fibrosis-Related Genes in the Crohn’s Disease and Ulcerative Colitis

In CD with fibrosis in the colon, the expression of genes *CXCL9*, *THBS2*, *MGP*, *PTPRC*, *CD52*, *GZMA* and *DCN* (ΔΔCq = 6.73, *p* < 0.001; ΔΔCq = 3.08, *p* < 0.001; ΔΔCq = 1.66, *p* < 0.001; ΔΔCq = 3.55, *p* < 0.001; ΔΔCq = 1.90, *p* < 0.001; ΔΔCq = 1.17, *p* = 0.03; ΔΔCq = 1.03, *p* < 0.001, respectively) was significantly up-regulated compared with normal colon, while the gene expression of *DPT* (ΔΔCq = −0.16, *p* = 0.91) was not significantly altered (Figure 3, Table 4). 

In CD with inflammation without fibrosis in the colon, *CXCL9*, *PTPRC* and *GZMA* (ΔΔCq = 2.56, *p* < 0.001; ΔΔCq = 1.46, *p* < 0.001; ΔΔCq = 4.33, *p* < 0.001, respectively) were significantly up-regulated compared with normal colon; *DPT* and *DCN* (ΔΔCq = −1.32, *p* < 0.001; ΔΔCq = −1.30, *p* < 0.001, respectively) were down-regulated; and the gene expression of *THBS2*, *MGP* and *CD52* (ΔΔCq = 0.25, *p* = 0.57; ΔΔCq = −0.47, *p* = 0.25; ΔΔCq = −0.28, *p* = 0.43, respectively) was not significantly altered. 

In UC, *CXCL9*, *CD52* and *GZMA* (ΔΔCq = 2.55, *p* < 0.001; ΔΔCq = 1.69, *p* < 0.001; ΔΔCq = 5.78, *p* < 0.001, respectively) were significantly up-regulated compared with normal colon, while the expression of *DPT* and *DCN* (ΔΔCq = −3.13, *p* < 0.001; ΔΔCq = −0.49, *p* = 0.02, respectively) was significantly down-regulated and the gene expression of *THBS2*, *MGP* and *PTPRC* (ΔΔCq = 0.66, *p* = 0.05; ΔΔCq = −0.66, *p* = 0.134; ΔΔCq = 0.58, *p* = 0.09, respectively) was not significantly altered. 

In CD with fibrosis, compared with UC, the expression of *CXCL9*, *THBS2*, *MGP*, *PTPRC*, *DPT* and *DCN* was significantly up-regulated (ΔΔCq = 4.18, *p* < 0.001; ΔΔCq = 2.41, *p* < 0.001; ΔΔCq = 2.32, *p* < 0.001; ΔΔCq = 2.97, *p* < 0.001; ΔΔCq = 2.97, *p* < 0.001; ΔΔCq = 1.52, *p* < 0.001, respectively), the expression of *GZMA* was significantly down-regulated (ΔΔCq = −4.62, *p* < 0.001), while *CD52* expression (ΔΔCq = 0.21, *p* = 0.305) was not significantly different (Figure 4, Table 5). 

In CD with fibrosis, compared with CD without fibrosis, the expression of *CXCL9*, *THBS2*, *MGP*, *PTPRC*, *CD52*, *DPT* and *DCN* was significantly up-regulated (ΔΔCq = 4.17, *p* < 0.001; ΔΔCq = 2.83, *p* < 0.001; ΔΔCq = 2.13, *p* < 0.001; ΔΔCq = 2.09, *p* < 0.001; ΔΔCq = 2.18, *p* < 0.001; ΔΔCq = 1.20, *p* < 0.01; ΔΔCq = 2.17, *p* < 0.001, respectively), while the expression of *GZMA* was significantly down-regulated (ΔΔCq = −3.16, *p* < 0.001).

Gene expression correlation within each sample group was calculated and is presented in Appendix A.

## 4. Discussion

Using a bioinformatics approach, we identified 57 genes with a significantly altered expression in liver and kidney fibrosis in comparison with the corresponding normal tissue. Protein–protein interaction network analysis revealed two groups of linked proteins involved in extracellular matrix organization and the immune response. From the two groups, we identified *CXCL9, THBS2, MGP, PTPRC, CD52, GZMA, DPT* and *DCN* as genes with the most significantly altered expression in both kidney and liver fibrosis, suggesting that they may be involved in the common pathogenetic mechanisms of fibrosis. The analysis of gene expression in our experimental setting with biopsy samples of advanced liver and kidney fibrosis showed the predicted changes in gene expression for the majority of the selected genes. Gene expression changes in biopsies matched the bioinformatics study, since the expression of all genes reaching significant levels in the bioinformatics study was also up-regulated in biopsies. Therefore, we believe that the results of the selected genes’ analysis in the kidney and liver fibrosis strongly support the results and conclusions of the bioinformatics part of our study.

Next, we analyzed the same genes in CD and UC. In CD with fibrosis compared with normal colon, seven of the eight studied genes were up-regulated, as was predicted in our study, while *DPT* expression was not significantly altered. These results show strong concordance with the bioinformatics results and with the results of our expression analysis of samples of kidney and liver fibrosis. Notably, *THBS2, MGP, PTPRC* and *CD52* were significantly up-regulated in all fibrosis groups in our study. These results indicate that fibrosis in CD follows the same pathways as in other organs and might be responsive to similar therapeutic interventions.

Comparing different samples of IBD, CD with fibrosis showed significant differences in comparison to both UC and CD without fibrosis, indicating more similarities of CD without fibrosis to UC than to CD with fibrosis. While some of the studied genes were significantly up-regulated in CD without fibrosis or UC, they showed much higher up-regulation in CD with fibrosis, which was still significantly higher in all studied genes except for genes *CD52* and *GZMA*. The increased expression of certain fibrosis-related genes in UC and CD without fibrosis could suggest the activation of pro-fibrotic pathways early in the disease process, that are then even more up-regulated in frank fibrosis or could reflect fibrosis limited to the submucosa.

At least seven of the eight identified genes have been studied for their involvement in fibrosis in individual organs; however, most have not been linked to fibrosis in all organs included in our study. To our knowledge, only *CXCL9, PTPRC* and *DCN* have been studied in association with IBD until now [22,23,24]. Similarly, a recent genetic study on stricturing CD showed the involvement of collagens and *THBS2*, which were also identified in our bioinformatics analysis [25].

We believe that these results validate our approach, showing high specificity for fibrosis-related genes, most of which have yet not been studied in IBD, giving new potential targets for research and for treating intestinal fibrosis. 

CXCL9 is a cytokine, that binds to the CXCR3 receptor. It mainly has a role in immune cell migration, activation and differentiation [26]. In addition to its function in the immune system, CXCL9 has been shown to activate cardiac fibroblasts [27] and intestinal myofibroblasts [28]. Studies in the lung have identified it to regulate fibroblast activation and extracellular matrix deposition [29,30]. It has already been studied in IBD for its role in the immune response [22], where there is some evidence that it could drive inflammation in patients that have become unresponsive to anti-TNF therapy [31]; however, there are few studies on its role in intestinal fibrosis. CXCL9 seems a particularly promising target for additional research because several antagonists for its receptor, CXCR3, have been designed that could potentially function as a treatment that targets both inflammation and fibrosis in IBD [22]. In our study, *CXCL9* was up-regulated in both kidney and liver fibrosis. It also showed up-regulation in all IBD samples, with a much higher level in CD with fibrosis, suggesting a role of CXCL9 in both inflammation and fibrosis, but further studies are needed to clarify its role in more detail.

*THBS2* (thrombospondin-2) encodes a protein that mediates cell–cell and cell–matrix interactions. It has been associated with fibrosis in the liver [32], lungs [33] and kidneys [34]. In the liver, its expression correlates with the degree of fibrosis and it has been shown to activate and promote the proliferation of hepatic stellate cells that mediate fibrosis in the liver [32]. In our study, *THBS2* showed up-regulation in all fibrosis groups and was not significantly altered in UC and CD without fibrosis.

MGP (Matrix Gla protein) is a vitamin K-dependent matrix protein with a high affinity for calcium ions. It is best known for its role in vascular mineralization and bone organization [35]. In association with fibrosis it has been shown to correlate with the progression of kidney fibrosis [36]. It has also been proposed that vitamin K might slow the progression of idiopathic pulmonary fibrosis through the activation of MGP [35]. In our study, *MGP* showed up-regulation in all fibrosis groups and was not significantly altered in UC and CD without fibrosis.

*PTPRC* encodes protein tyrosine phosphatase, receptor type C, better known as CD45 antigen; it is a member of the protein tyrosine phosphatase family and is found on all differentiated hematopoietic cells (except erythrocytes and plasma cells) as well as on fibrocytes [37]. Fibrocytes are circulating cells originating from the bone marrow that have been shown to have a key role in fibrosis as a possible precursor of myofibroblasts in many different organs, such as the lung, kidney and liver [38,39,40], as well as the bowel in IBD [24]. Fibrocytes are recognized by a CD34+ CD45+, Col1+ phenotype [39]. In our study, we cannot distinguish between *PTPRC* expression up-regulation caused by an influx of PTPRC+ fibrocytes or PTPRC+ leukocytes. In CD fibrosis, it is most likely a combination of *PTPRC* up-regulation by fibrosis-generating fibrocytes and inflammatory leukocytes. In the liver and kidney fibrosis, there is less inflammation, which could indicate a greater fibrosis-specific signal. However, due to a low baseline inflammatory cell content in normal liver and kidney tissue, even a small increase in inflammatory cells could cause a significant change in expression. In our study, *PTPRC* was up-regulated in kidney and liver fibrosis in CD with fibrosis and in CD without fibrosis; it was not significantly altered in UC. The up-regulation of *PTPRC* in CD without fibrosis could be a signal of early fibrosis before it becomes apparent or a CD-specific finding in IBD.

CD52 has a role in leukocyte migration, proliferation and the function of regulatory T lymphocytes [41]. There are limited data on CD52 in fibrosis; we found a single study that showed an increased CD52 expression in alveolar macrophages in interstitial lung disease [42]. In our study, *CD52* was up-regulated in all fibrosis groups and also in CD without fibrosis, but it was not significantly altered in UC. 

*GZMA* encodes the granzyme A protein, which is a serine protease and a component of cytotoxic T lymphocyte granules. In addition to its cytotoxic role, it has been shown that granzymes have a role in tissue remodeling and fibrosis [43]. This has been shown by demonstrating increased levels of granzymes in various fibrotic states [43,44,45] and by showing that genetic deficiency of granzyme B has a protective effect on cardiac fibrosis [46]. However, most research has been conducted on granzyme B, a close relative of granzyme A. In our study, *GZMA* was not significantly altered in liver and kidney fibrosis. It was up-regulated in CD with fibrosis and even more in CD without fibrosis and UC, which seems more consistent with a role in inflammation as part of a T cell response and not in fibrosis. Notably, *GZMA* was up-regulated in the bioinformatics analysis, where the samples of fibrosis were from kidney allograft rejection and hepatitis B liver fibrosis, both diseases that typically feature inflammation, while it was not altered in our samples which were specifically selected to minimize inflammation.

*DPT* encodes dermatopontin, which is an extracellular matrix protein that promotes collagen fibril formation and cell adhesion [47]. It has been shown to increase the availability of active TFG-β [47], which is known to be one of the most important cytokines in fibrosis, including intestinal fibrosis [10]. It has also been shown to be involved in several fibrotic states [48,49], including liver cirrhosis, where its expression increases in fibrosis and decreases after the reversal of fibrosis [50]. In our study, *DPT* was up-regulated in liver fibrosis and was not significantly altered in kidney fibrosis and CD with fibrosis. It was down-regulated in UC and CD without fibrosis in comparison with both normal colon and CD with fibrosis.

*DCN* encodes decorin, an extracellular matrix protein that has a role in collagen fibril formation [51]. More importantly, decorin interacts with several growth factors and receptors, including TGF-β, which is the most important pro-fibrotic cytokine. Decorin forms complexes with free TGF-B, thus inhibiting TGF-β signaling, and reduces scar formation [51]. TGF-β on the other hand stimulates decorin expression, forming a negative feedback loop on TGF-β signaling [52]. Decorin has been shown to compete with dermatopontin for TGF-β binding, with enhanced TGF-β binding when decorin and dermatopontin form a complex [47,53]. The full extent of decorin–dermatopontin–TGF-β interactions is still unknown, while both decorin and dermatopontin have an important role in collagen fibril formation [47], indicating complex pro- and antifibrotic interactions that could be important for the progression or cessation of fibrosis. Decorin has been shown to be increased in fibrosis in various organs, including the skin, kidney and lung. The application of exogenous decorin decreases fibrosis in lung [54] and kidney [55] fibrosis models. An IBD mouse model showed an increased expression of decorin in affected intestinal tissue [23]. In our study, *DCN* was up-regulated in kidney fibrosis and it was not significantly altered in liver fibrosis. In IBD, it was up-regulated in CD with fibrosis and down-regulated in UC and CD without fibrosis. 

Our study has some limitations. It is a purely observational study on biopsy cases with established fibrosis. Certain genes important in the process of fibrosis could have different expression during the formation of fibrosis compared with established fibrosis. We also cannot rule out a possible effect of treatment on gene expression. Because IBD is a disease that causes significant morbidity in young patients, leading to bowel resection much earlier than in other diseases, there is a large age difference between the IBD groups and the normal bowel group, which could be a reason for some changes in gene expression. Due to the difficulty of finding samples of fibrosis without other significantly confounding pathologies, the numbers of cases in all groups are relatively small. Due to the observational nature of the study, we cannot confirm the importance of the identified genes in fibrosis; however, we think this provides a good starting point for further study.

## 5. Conclusions

We found similar changes in the expression of selected fibrosis-related genes in liver and kidney fibrosis, as well as in fibrosis in CD. Some of these genes were already active in CD with inflammation without significant fibrosis, but to a lesser degree, suggesting an early activation of profibrotic pathways or overlapping function in both fibrosis and inflammation.

Our results provide further evidence that fibrosis in IBD might share some common mechanisms with fibrosis in other organs, supporting the hypothesis that fibrosis is the final common pathway in diseases of various organs. However, our results also indicate that there may be some differences in the mechanisms of fibrosis development in the colon. 

## Figures and Tables

**Figure 1 cells-11-00314-f001:**
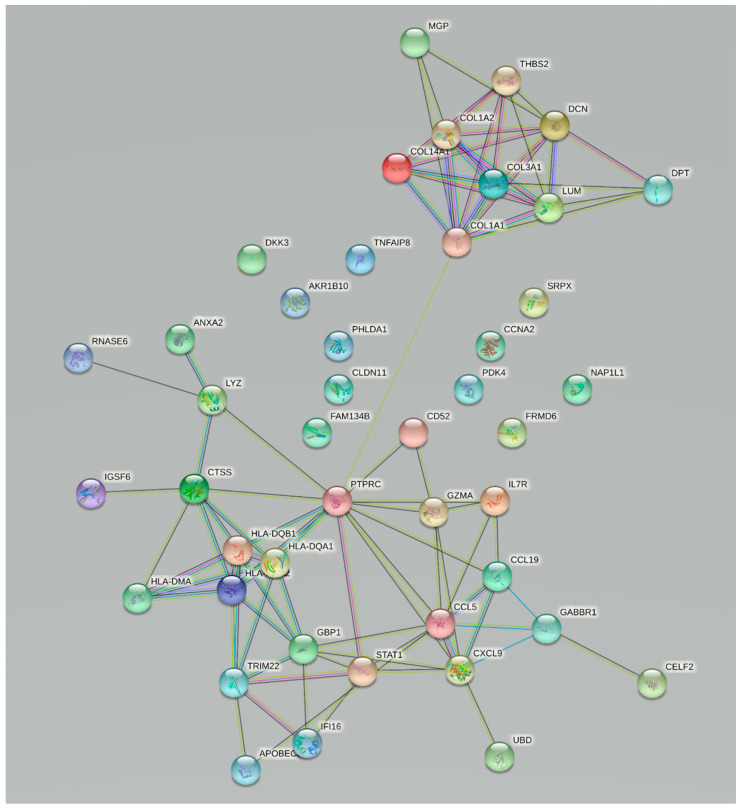
Protein–protein interaction network of intersection of significantly deregulated genes between kidney and liver fibrosis.

**Figure 2 cells-11-00314-f002:**
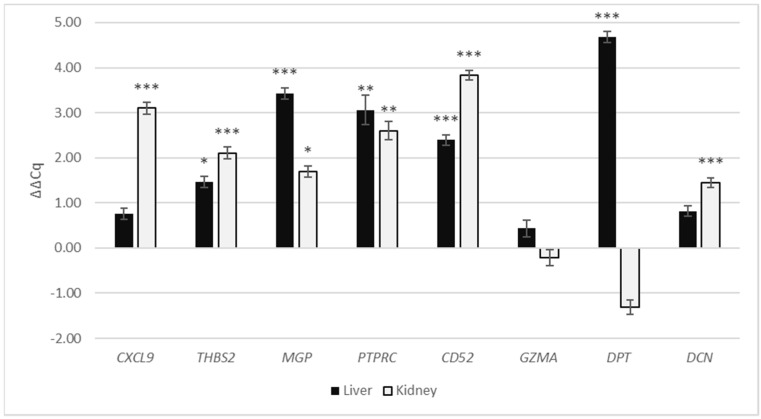
Gene expression of *CXCL9, THBS2, MGP, PTPRC, CD52, GZMA, DPT* and *DCN* in fibrotic kidney and liver expressed as ΔΔCq relative to normal kidney and liver, respectively. Legend: * *p* ≤ 0.05; ** *p* ≤ 0.01; *** *p* ≤ 0.001.

**Figure 3 cells-11-00314-f003:**
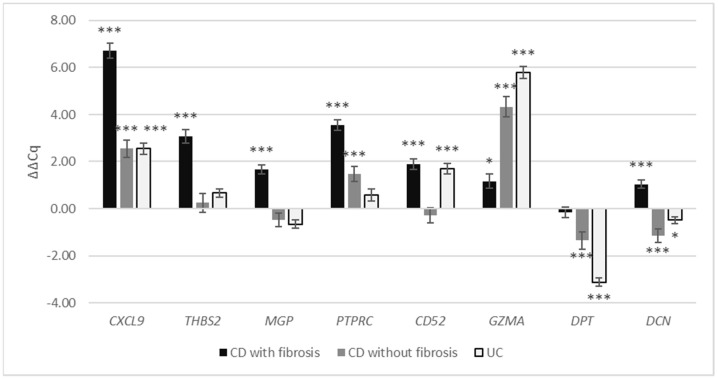
Gene expression of *CXCL9*, *THBS2*, *MGP*, *PTPRC*, *CD52*, *GZMA*, *DPT* and *DCN* in colonic CD with fibrosis, colonic CD without fibrosis and UC expressed as ΔΔCq relative to normal colon. Legend: * *p* ≤ 0.05; *** *p* ≤ 0.001; CD, Crohn’s disease; UC, ulcerative colitis.

**Figure 4 cells-11-00314-f004:**
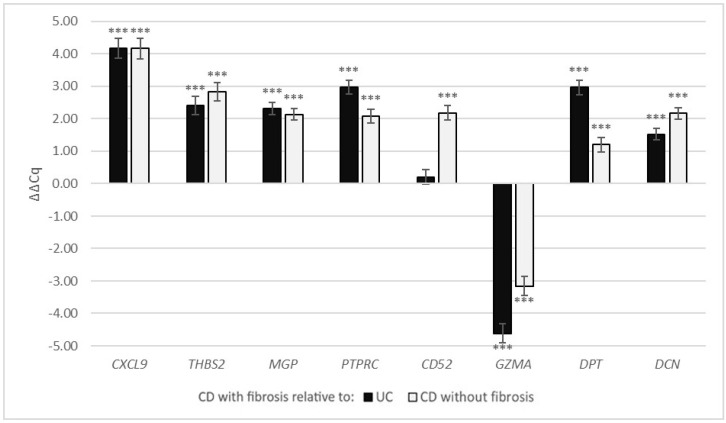
Gene expression of *CXCL9*, *THBS2*, *MGP*, *PTPRC*, *CD52*, *GZMA*, *DPT* and *DCN* in colonic CD with fibrosis as ΔΔCq relative to colonic CD without fibrosis and UC. Legend: *** *p* ≤ 0.001; CD, Crohn’s disease; UC, ulcerative colitis.

**Table 1 cells-11-00314-t001:** Differences in expression of six chosen genes between normal and fibrotic tissue expressed as fold change in kidney and liver tissue. Legend: logFC, logarithm of fold change.

	Kidney		Liver	
Gene	logFC	*p*-Value	logFC	*p*-Value
*PTPRC*	2.96	1.30 × 10^−5^	1.28	5.78 × 10^−8^
*CXCL9*	2.54	4.06 × 10^−4^	1.95	2.75 × 10^−8^
*CD52*	2.53	3.39 × 10^−6^	1.33	4.58 × 10^−9^
*GZMA*	1.89	2.29 × 10^−5^	1.13	2.48 × 10^−8^
*DCN*	1.64	7.75 × 10^−5^	1.11	2.58 × 10^−13^
*THBS2*	1.47	4.57 × 10^−5^	1.90	1.58 × 10^−15^
*DPT*	1.33	2.32 × 10^−3^	1.26	4.09 × 10^−8^
*MGP*	1.01	9.31 × 10^−3^	1.89	8.65 × 10^−12^

**Table 2 cells-11-00314-t002:** Basic patients’ demographic data.

	Number of Patients	Age in Years (St. Dev)	Gender (M:F)
Normal liver	7	59.2 (21.0)	2:1 ^1^
Liver fibrosis	10	62.7 (10.9)	7:3
Normal kidney	9	34.6 (18.7)	2:7
Kidney fibrosis	11	65.8 (10.9)	9:2
Normal bowel	11	78.9 (8.7)	4:7
Crohn’s disease with fibrosis	14	36.1 (14.4)	9:5
Crohn’s disease without fibrosis	14	35.8 (15.9)	9:5
Ulcerative colitis	10	42.2 (20.5)	4:6

^1^ 4 patients in the control liver group had unknown gender because the samples were from pre-transplantation screening biopsies of potential donor livers without patient data.

**Table 3 cells-11-00314-t003:** Gene expression of *CXCL9, THBS2, MGP, PTPRC, CD52, GZMA, DPT* and *DCN* in fibrotic kidney and liver expressed as ΔΔCq relative to normal kidney and liver, respectively.

	Liver		Kidney	
Gene	ΔΔCq	*p*-Value	ΔΔCq	*p*-Value
*CXCL9*	0.76	0.206	3.10	<0.001
*THBS2*	1.46	0.031	2.11	0.001
*MGP*	3.43	0.001	1.70	0.012
*PTPRC*	3.06	0.002	2.60	0.002
*CD52*	2.39	<0.001	3.83	<0.001
*GZMA*	0.43	0.310	−0.21	0.635
*DPT*	4.68	<0.001	−1.31	0.080
*DCN*	0.81	0.056	1.45	<0.001

**Table 4 cells-11-00314-t004:** Gene expression of *CXCL9, THBS2, MGP, PTPRC, CD52, GZMA, DPT* and *DCN* in colonic CD with fibrosis, colonic CD without fibrosis and UC expressed as ΔΔCq relative to normal colon. Legend: CD, Crohn’s disease; UC, ulcerative colitis.

	CD with Fibrosis		CD without Fibrosis		UC	
Gene	ΔΔCq	*p*-Value	ΔΔCq	*p*-Value	ΔΔCq	*p*-Value
*CXCL9*	6.73	<0.001	2.56	0.001	2.55	<0.001
*THBS2*	3.08	<0.001	0.25	0.574	0.66	0.051
*MGP*	1.66	<0.001	−0.47	0.252	−0.66	0.134
*PTPRC*	3.55	<0.001	1.46	<0.001	0.58	0.091
*CD52*	1.90	<0.001	−0.28	0.432	1.69	<0.001
*GZMA*	1.17	0.030	4.33	<0.001	5.78	<0.001
*DPT*	−0.16	0.905	−1.36	0.001	−3.13	<0.001
*DCN*	1.03	<0.001	−1.14	0.001	−0.49	0.019

**Table 5 cells-11-00314-t005:** Gene expression of *CXCL9*, *THBS2*, *MGP*, *PTPRC*, *CD52*, *GZMA*, *DPT* and *DCN* in colonic CD with fibrosis as ΔΔCq relative to colonic CD without fibrosis and UC. Legend: CD, Crohn’s disease; UC, ulcerative colitis.

	UC		CD without Fibrosis	
Gene	ΔΔCq	*p*-Value	ΔΔCq	*p*-Value
*CXCL9*	4.18	<0.001	4.17	<0.001
*THBS2*	2.41	<0.001	2.83	<0.001
*MGP*	2.32	<0.001	2.13	<0.001
*PTPRC*	2.97	<0.001	2.09	<0.001
*CD52*	0.21	0.305	2.18	<0.001
*GZMA*	−4.62	<0.001	−3.16	<0.001
*DPT*	2.97	<0.001	1.20	0.001
*DCN*	1.52	<0.001	2.17	<0.001

## Data Availability

The data underlying this article are available in the article and in its online Appendix A.

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
