# Peer review of "Expression of Fibrosis-Related Genes in Liver and Kidney Fibrosis in Comparison to Inflammatory Bowel Diseases"

_cells, 2022, doi:10.3390/cells11030314_

Round 1

Reviewer 1 Report

The article by Jerala et al., discussed about the differential gene expression common between liver, kidney and IBD. The authors reported similarity in 8 different genes (CXL9, THBS2, MGP, PTPRC, CD52, 11 GZMA, DPT and DCN). Since the new data generated by the authors are the fibrosis genes for Crohn’s  disease (CD) with and without fibrosis and ulcerative colitis (UC) some more focus has to be given to that topic. Discussing more on the differentially expressed genes in related to intestinal fibrosis will be beneficial.The article may consider reviewing some important published articles in relation to this and possibly improve the article.

In general, the following improvements are suggested

  1. Biglycan is also a member of the proteoglycan family similar to decorin. Biglycan is reported to be regulated by TGF-β (stimulates biglycan production). Do you see the same trend in IBD fibrosis?
  2. Is there any differential expression of genes related to ECM/migration (Collagens) were present in CD with fibrosis (any of the genes discussed in Feng Wu, Shukti Chakravarti,2007)
  3. Gu et al., 2021 reported 150 intestinal fibrosis genes overlap with other eight fibrotic diseases. Did you compare your results with that?
  4. DPT is also reported to compete with decorin for binding with TGF-β. Can you add some explanation for this?
  5. What about granzyme B? GrB/perforin pathway involvement in allograft rejection has been suggested. (Choy et al.,2010)
  6. Please add some discussion about PTPRC in kidney and liver?
  7. Is it possible to include the IHC images of the tissue samples?
  8. Add a table showing the differential expression of genes corresponding to Figure 3,4.
  9. Is there any common DGE between kidney and CD with fibrosis as well as liver and CD with fibrosis? Also is there any DGE only in CD with fibrosis but not in liver and kidney?
  10. Is there any significant difference between matrix metalloproteases (MMPs) in Crohn’s disease and ulcerative colitis with normal colon? A discussion about this will be beneficial.
  11. Including the images of some of the pathways discussed with the figures will also improve the manuscript.

Author Response

We appreciate your critical and valuable comments which have contributed to the quality of the paper and hope that the paper is now acceptable for publication.

Reviewer 2 Report

Jerala et al. investigate the expression of fibrosis-related genes in fibrosis of the liver and the kidneys in the context of IBD, which is a very interesting topic. This is useful research exploring genes with potential role in fibrosis. The conclusions cannot be very strong because of the limitations and the scope of research, but they are nevertheless useful, offering validation and additional certainty regarding some crucial results. I have only minor comments.

Abstract “Gene expression omnibus” – should it not be Gene Expression Omnibus? Please check what is the correct spelling.

Abstract “in various organs 19 diseases” – please consider “in diseases of various organs”

In 2.1. it should be added that liver fibrosis resulted from HBV infection and kidney fibrosis was related to transplantation. This provides important context for the source of re-analyzed data.

It would be interesting to know what was the starting number of genes and what was the number of genes left in analysis after filtering for insufficient intensity < 100. Was limma run on data from kidney and liver together (if so, was batch-correction performed?) – or was it run separately on the kidney data, the liver data, and then the results were overlapped to investigate key similarities?

It does not necessitate further discussion or changes to the manuscript but in my experience using STRING with more stringent settings sometimes also yields interesting results. But the default settings should be fine for the purpose of this study, where many of the genes were still left out of the network and some connections were weak.

Do you think age could be a confounding factor?

The design is very interesting, with very diverse sources of samples. This is a strength but also a challenge because the samples may not be directly comparable.

Of note, IPO8 (here used as a reference gene) may have importance in IBD and may be involved in TGF-beta and fibrosis, and also Lorey-Dietz syndrome. In my view this is not the best housekeeping gene for the study of fibrosis. Please see https://pubmed.ncbi.nlm.nih.gov/34010604/  But little can now be done. This should be, however, discussed as a limitation. Luckily, there was another reference gene. I am aware of studies indicating that IPO8 is a very good reference gene, I imagine this was the basis for the choice made by the authors. Yet, I checked on STRING that even adding 20 interactors there is no link between the 8 genes selected for qPCR analyses and IPO8.

Assessment in duplicate should be sufficient for this study, given the sample size.

Are p values given in the supplementary table raw p-values or FDR-corrected p-values? Looking at the table it is clear there is much more overexpression than underexpression. Only mitochondrial Pyruvate Dehydrogenase Kinase 4 is significantly downregulated. It is influenced by the retinoic acid. Its downregulation may be a protective reaction against fibrosis – please look at this study https://journals.physiology.org/doi/full/10.1152/ajpheart.00870.2007 - “Although no overt cardiomyopathy was observed in the PDK4 transgenic mice, introduction of the PDK4 transgene into mice expressing a constitutively active form of the phosphatase calcineurin, which causes cardiac hypertrophy, caused cardiomyocyte fibrosis and a striking increase in mortality.” – this suggests that some of the observed changes may actually protect against fibrosis and not cause it.

The choice of genes for further investigation is somewhat arbitrary but there is a reason for the choice that was made: some of the genes are related to ECM and others to immunity. I think STAT1 could be a good candidate in the latter group, also because of STAT involvement in TGFbeta pathway.

Looking at the network – it is interesting that TRIM22 is related to very early-onset inflammatory bowel disease https://www.sciencedirect.com/science/article/pii/S0016508516001232

The presence of LYZ would implicate innate immunity.

The sample size is small but sufficient when considered that data from larger studies were re-analyzed in preparation for this study.

This does not need discussion in the manuscript – why did the authors did not simply choose the genes that had the highest logFC in both liver and kidney? Were the eight genes checked for correlations between themselves, were they independent of each other? Were COL1A1 and similar genes considered uninteresting (they had high logFC in both) as just reflecting fibrosis?

Figure 3 it may be interesting, if feasible, to show differences between the groups (not only relative to normal colon). In Figure 3 caption it would be useful to underscore that it was colonic CD tissue.

It is interesting to see that CD52 was higher in fibrotic CD tissue and UC but not CD without fibrosis; this is a marker of B and T cells. Maybe the sample size is insufficient.

Is it possible that lower expression of GZMA favorizes fibrosis (looking at Figure 3 and knowing that it is overexpressed in fibrosis in kidney and liver, maybe as a protective reaction to ongoing processes) (similar hypothesis for DPT?)  ?

The detailed results of qPCR are hard to read (probably better presented in a table, not necessary) but luckily there are figures.

Figure 4 – there should be some kind of information on the figure. E.g., as there is a small legend at the bottom (“UC” and “CD without fibrosis”) it could be added to the left of it: “CD with fibrosis relative to: ….” And then it would be easier to interpret intuitively. Of course, it is described under the figure, but the figure itself is not intuitive – a small change may be very helpful.

Line 221 “all gene expressions” – „the expression of all genes”?

Line 223 “strongly supports”

One of the main difficulties is to understand whether CXCL9 overexpression has a role in fibrosis or protection against fibrosis. One could speculate it is the first, but it may be more complex.

Lines 288-289 “Up-regulation of PTPRC in CD without fibrosis could be a signal of early fibrosis 288 before it becomes apparent or a CD specific finding in IBD” – or it could suggest that it has a protective activity?

Lines 301-303 Yes, this is also a good point – it may just be involved in inflammation.

The discussion of the investigated genes and their functions is interesting.

The limitations section is frank and touches upon the main difficulties.

Conclusion line 345 “final common pathway in organ failure in diseases of various organs” – this study did not investigate organ failure? Maybe “final common pathway in diseases of various origin” would be sufficient, or a similar statement?

The end of conclusion – “probably related to the specificity of the bowel wall structure and function” – I have the impression this quite suddenly appears without much prior discussion or motivation. I would suggest that the conclusion is strictly based on the results, very conservative.

It is a positive aspect that the work uses bioinformatics to target qPCR analyses, it is a productive and cost-efficient approach.

Bridging a gap between fibrosis of different origin and IBD is a creative approach.

Author Response

(The authors gave the same response as above.)

Round 2

Reviewer 1 Report

The author addressed all the comments and the article is acceptable in the current form.